# IMITATE OPTIMAL POLICY: PREVAIL AND INDUCE ACTION COLLAPSE IN POLICY GRADIENT

## ABSTRACT

Policy gradient (PG) methods in reinforcement learning frequently utilize deep neural networks (DNNs) to learn a shared backbone of feature representations used to compute likelihoods in an action selection layer. Numerous studies have been conducted on the convergence and global optima of policy networks, but few have analyzed representational structures of those underlying networks. While training an optimal policy DNN, we observed that under certain constraints, a gentle structure resembling neural collapse – which we refer to as *Action Collapse* (AC) – emerges. This suggests that 1) the state-action activations (i.e. last-layer features) sharing the same optimal actions collapse towards those optimal actions' respective mean activations; 2) the variability of activations sharing the same optimal actions converges to zero; 3) the weights of action selection layer and the mean activations collapse to a simplex equiangular tight frame (ETF). Our early work showed those aforementioned constraints to be necessary for these observations. Since the collapsed ETF of optimal policy DNNs maximally separates the pair-wise angles of all actions in the state-action space, we naturally raise a question: *can we learn an optimal policy using an ETF structure as a (fixed) target configuration in the action selection layer?* Our analytical proof shows that learning activations with a fixed ETF as action selection layer naturally leads to the Action Collapse. We thus propose the *Action Collapse Policy Gradient* (ACPG) method, which accordingly affixes a synthetic ETF as our action selection layer. ACPG induces the policy DNN to produce such an ideal configuration in the action selection layer while remaining optimal. Our experiments across various discrete Gym environments demonstrate that our technique can be integrated into any discrete PG methods and lead to favorable reward improvements more quickly and robustly.

## 1 INTRODUCTION

Reinforcement learning (RL) involves training agents to interact with environments by selecting actions aiming to maximize expected cumulative rewards or "returns". Classical methods such as Q-learning (Sutton, 1984) and SARSA (Watkins, 1989) do this by learning a accompanying value function. In contrast, "policy gradient" (PG) algorithms (Sutton et al., 1999) directly optimize the action selection layer [1] to maximize return by modeling action probabilities from state features. Modern PG methods use deep neural networks (DNNs) to estimate these probabilities.

Recent breakthroughs in artificial intelligence—such as GPT-4 (Peng et al., 2023)—have been driven by large language models (LLMs), with much of their later success attributed to RL. Reinforcement learning from human feedback (Askell et al., 2021; Sriyash et al., 2024; Dai et al., 2024) introduced a new paradigm of LLM post-training, many examples of which incorporate PG, such as TRPO (Silver et al., 2014; Schulman et al., 2015), PPO (Schulman et al., 2017), and DPO (Rafailov et al., 2023).

Despite these empirical successes in RL, a substantial gap persists between the theory and practice of RL. Theoretical research in RL has primarily focused on the convergence of RL algorithms (Gaur et al., 2023; Fan et al., 2020; Asadi et al., 2023; Jin et al., 2018b) and understanding how to effectively train an optimal policy (Xiong et al., 2022b) [2]. Other studies attempt to tackle the exploration–exploitation

---

[1] The last layer of PG DNNs that computes the likelihoods for each action, a.k.a. the action classifier/head.
[2] The optimal policy is defined as a policy which outputs the optimal action and achieves the highest return.

dilemma which requires policy networks to strike a balance between exploring previously unvisited states in the experience replay buffer (Lin, 1992) and exploiting existing policy efficiently. Recent works have motivated policies explicitly or implicitly (Thrun, 1992), to take exploratory actions to discover new outcomes, even in the absence of immediate rewards or theoretical guarantees of convergence (Xiong et al., 2022a; Strehl & Littman, 2005). However, it is still unclear how an optimal policy is to be derived and what underlying DNN structures achieve such optimality.

A set of phenomena known as "Neural Collapse" (NC) (Papyan et al., 2020), characterizes the structure of optimal classification DNNs trained to convergence using canonical objectives such as cross-entropy (CE) (Papyan et al., 2020) or mean square error (MSE) (Han et al., 2021). Several works have demonstrated that 1) the last-layer features of data samples to collapse to their respective class means; 2) those means construct a simplex equiangular tight frame (ETF) (Strohmer & Heath Jr, 2003); 3) classifier weights also converge to this ETF; and 4) classification decisions converge to a near class centre classifier. Beyond its elegant geometric symmetry, the theoretical breakthrough of NC has proven valuable in inspiring new training methods for classification models, particularly under conditions of data imbalance and scarcity (Xie et al., 2023; Yang et al., 2023).

PG methods share key similarities with classification tasks. For instance, PG is applied to the classification of positive and unlabeled data, where a policy DNN is trained to infer data labels (Li et al., 2019). A notable example is Goal-Conditioned Supervised Learning, which reformulated RL states and target rewards as input and the corresponding action as label (Ghosh et al., 2021). Based on the existence of NC and conceptual parallels between RL and classification, we aim to investigate the structures of optimal DNNs for PG. From the perspective of cost function formulation, PG is essentially CE weighted by a Q-value function (Sutton et al., 1999). Therefore, we empirically assess whether phenomena analogous to NC emerge during PG training in several discrete RL environments.

A similar phenomenon—which we refer to as *Action Collapse*—is observed in our ideal discrete RL environments under certain constraints. However, real-world RL environments train DNNs within a limited time frame (Zhang et al., 2019), resulting in data scarcity. RL samples state-action sets from the environment (Sutton et al., 1999), which may exhibit significant imbalance across different states due possibly to greedy exploitation and incomplete knowledge of the environment, and usually to the natural imbalance in the task at hand. Since Action Collapse is rarely observed in real-world problems, we further consider: **can we learn an optimal policy using an ETF structure as a (fixed) target configuration in the action selection layer for realistic discrete RL problems where ideal condition cannot be assumed?**

In this paper, we propose Action Collapse Policy Gradient (ACPG), which fixes the action selection layer as a synthetic simplex ETF. Our experiments demonstrate that ACPG achieves higher returns than previous PG methods, with faster and more robust convergence. In addition, our theoretical analysis shows that even in the absence of idealized empirical constraints, the activation means still collapse to a ETF under ACPG training. In other words, Action Collapse naturally develops in practical RL environments using our proposed technique. Main contributions of this work can be summarized as follows:

- To the best of our knowledge, this is the first study to empirically analyze the structure of optimal policy DNNs in fully explored discrete RL environments, in which we observed the Action Collapse phenomenon (under certain constraints). We note that Action Collapse may not naturally occur in more realistic environments, due to factors such as insufficient exploration, imbalanced sampling, or the natural imbalance of states/actions.

- To address this, we propose ACPG, which initializes action selection layer as a fixed simplex ETF structure. Accordingly, we analytically prove that Action Collapse optimality can still be induced in such realistic environments under ACPG.

- We validate ACPG across **10+** Gym environments, **4+** different PG methods. Our proposed method can be integrated with any discrete PG algorithms, and consistently achieves more stable, faster, and robust convergence, leading to significant performance improvements.

## 2 PRELIMINARIES

### 2.1 POLICY GRADIENT

The goal of PG is to find an optimal policy, which in our case is determined by a trained DNN parameterized by $\theta$. $\pi_\theta(a|s)$ represents the probability of choosing action $a$ regarding environment observation $s$ under policy $\pi_\theta$ [3]. The value from the reward function depends on the policy, and various algorithms (Schulman et al., 2015; 2017; Cobbe et al., 2020) can be applied to optimize $\theta$ to maximize the expected return value.

**Definition 1** (Reward/Objective Function of Policy Gradient Methods). *Normally an discrete RL environment contains a state set $\mathcal{S}$ and the discrete action space $A$, with $|A| = K$. We define the set as $\mathcal{S}_k = \{s : opt(s) = a_k\}$, where $opt(s)$ denotes the true optimal action for state $s$.*

*The reward function is defined as*

$$J(\theta) = \sum_{s \in \mathcal{S}} d^\pi(s) \sum_{a \in \mathcal{A}} \pi_\theta(a|s) \Psi^\pi, \tag{1}$$

*where $d^\pi(s)$ is the stationary distribution of the Markov chain induced by $\pi_\theta$ (i.e., the on-policy state distribution under $\pi_\theta$), with $d^\pi(s) = \lim_{t \to \infty} P(S_t = s|S_0, \pi)$. $\Psi$ denotes the "expected return" when action $a$ is chosen at state $s$ (Sutton et al., 1999).*

In RL settings, $\pi_\theta(a|s)$ represents a stable probability distribution over actions given a state $s$ when following policy $\pi_\theta$. The "expected return" $\Psi$ quantifies the future reward obtained by taking action $a$ in state $s$, which may vary across different algorithms. It may represent either 1) the remaining reward accumulation along a trajectory: $\sum_{t=\text{after } s,a}^\infty r_t$; or 2) the return following action $a_t$: $\sum_{t'=t}^\infty r_{t'}$, etc. For simplicity, we assume that a ground-truth value (optimal return) exists.

According to the PG loss function while optimizing $J(\theta)$, after the environment is fully explored and experiences for all $\mathcal{S}$ and $\mathcal{A}$ are collected, $J(\theta)(\mathbf{H})$ can be expressed as:

$$J(\theta)(\mathbf{H}) = \sum_{s \in \mathcal{S}} d^\pi(s) \cdot \log \frac{e^{h(s;\theta)^T w_{a^*}}}{\sum_{a \in \mathcal{A}} e^{h(s;\theta)^T w_a}} \Psi^\pi, \tag{2}$$

where $\mathbf{H} = \{h(s;\theta) : s \in \mathcal{S}\}$ and $h(s;\theta) \in \mathbb{R}^d$ is the activation, $\mathbf{W} = [w_1; \cdots; w_K] \in \mathbb{R}^{K \times d}$ is the action selection layer and $a^*$ is the optimal action predicted by the model.

### 2.2 NEURAL COLLAPSE

"Neural Collapse" (NC) was observed and formalized in classification tasks (Papyan et al., 2020), where feature maps converge to their within-class means, which together with the classifier vectors, collapse to the vertices of a simplex ETF during the terminal phase of training (TPT). See Appendix B for details.

**Definition 2** (Simplex Equiangular Tight Frame). *A collection of vectors $m_i \in \mathbb{R}^d$, $i = 1, 2, \cdots, K$, $d \geq K - 1$, are said to be a simplex ETF if:*

$$\mathbf{M} = \sqrt{\frac{K}{K-1}} \mathbf{U} \left( \mathbf{I}_K - \frac{1}{K} \mathbf{1}_K \mathbf{1}_K^T \right), \tag{3}$$

*where $\mathbf{M} = [m_1, \cdots, m_K] \in \mathbb{R}^{d \times K}$, $\mathbf{U} \in \mathbb{R}^{d \times K}$ is an orthogonal matrix satisfying $\mathbf{U}^T \mathbf{U} = \mathbf{I}_K$, $\mathbf{I}_K$ is the identity matrix, and $\mathbf{1}_K$ is the all-ones vector.*

All vectors in a simplex ETF have unit $\ell_2$ norm and the same pair-wise angle,

$$m_i^T m_j = \frac{K}{K-1} \delta_{i,j} - \frac{1}{K-1}, \forall i, j \in [1, K], \tag{4}$$

where $\delta_{i,j}$ equals to 1 when $i = j$ and 0 otherwise. The pair-wise cosine similarity $\frac{-1}{K-1}$ is the maximal equiangular separation of $K$ vectors in $\mathbb{R}^d$ (Papyan et al., 2020).

---

[3]The subscript for parameter $\theta$ is omitted for the policy $\pi_\theta$ when the policy is present in the superscript of other functions; for example, $d^\pi$ and $\Psi^\pi$ should be $d^{\pi_\theta}$ and $\Psi^{\pi_\theta}$ if written in full.

## 3    EMPIRICAL OBSERVATION

In this subsection, we present the main empirical evidence to show Action Collapse.

First, following Cliff Walking (Demin, 2009), we define a similar environment: the agent navigates a **2x4** grid, where two exits are consistently placed in the upward and downward directions of the central **2x2** region of the grid. The diagram is shown in Fig.1.(a). We apply the REINFORCE algorithm (Sutton et al., 1999) and estimate the returns via Monte-Carlo estimation (Sutton et al., 1999) using episode samples to update the policy parameter $\theta$. We constrain the Cliff Walking environment using the following three conditions:

**Condition 1.** *The state is fully explored and discrete, i.e., $max(|\mathcal{S}|) = 12$ in our cliff walking.*

**Condition 2.** *The cardinality of each state set $\mathcal{S}_k$, as defined in DEF.1, remains consistent across different actions $a_k$ in experience replay, such that $|\mathcal{S}_k| = |\mathcal{S}_{k'}| \, \forall k, k' \in K$. Specifically, for the four directional actions in the experience replay of our ideal cliff walking, $\mathcal{S}_1 = \mathcal{S}_2 = \mathcal{S}_3 = \mathcal{S}_4$.*

**Condition 3.** *The sampling of states across different actions selected by any policy is expected to be with equal probability when stationary, $\forall s, s' \in \mathcal{S} \, , \, d^\pi(s) = d^\pi(s')$.*

In addition to our Ideal Cliff Walking environment, the metrics of Action Collapse in three other Gym environments, Discrete-Car-Racing (Car-Racing) [4], Breakout-V5 (Breakout) and ALE/Pong-V5 (Pong) (Brockman et al., 2016) are also investigated.

**Moment of policy DNNs convergence.** During training, we periodically snapshot the parameters of the policy DNN at designated epochs. Given all unique states observed from an environment, we extract both the activations of the action selection layer and the corresponding weight vectors.

Let $h(s; \theta)$ — henceforth shortened as $h$ — denote the state-action activation. We compute the global activation mean $h_G \in \mathbb{R}^d$: $h_G \triangleq Avg_{s \in \mathcal{S}}\{h_s\}$, where $s$ and $\mathcal{S}$ are defined in Sec.2. For each action $a_k, k \in 1, \ldots, K$, we define the mean activations of distinct states that share $a_k$ as the optimal action: $h_k \triangleq Avg_{i=1}^{n_k}\{h_{i,k}\}, \quad k = 1, \ldots, K$, where $n_k$ is the number of distinct state-action activations that share $a_k$ as the optimal action. Similarly, $w_k$ is weight of action selection layer of the $k^{th}$ action. To measure angularity, given two distinct action indexes $k, k'$, we denote $\cos_h(k, k') = \langle h_k - h_G, h_{k'} - h_G \rangle / (\|h_k - h_G\|_2 \|h_{k'} - h_G\|_2)$ and $\cos_w(k, k') = \langle w_k, w_{k'} \rangle / (\|w_k\|_2 \|w_{k'}\|_2)$.

**Results and Discussion.** Our main findings and results are presented in Fig.1. Action Collapse is clearly observed in **Ideal Cliff Walking** and **Car-Racing**, but not in real-world RL environments such as **Breakout** and **Pong**. In **Ideal Cliff Walking**, the near-zero values of equinorm, equiangularity, maximal-angle equiangularity, and angular variation indicate that both the state-action activations [5] and the action selection layer weights form a simplex ETF. Based on empirical observations, we propose Action Collapse as following:

$$\mathbf{M}^* := \sqrt{\frac{K}{K-1}} \mathbf{U} \left( \mathbf{I}_K - \frac{1}{K} \mathbf{1}_K \mathbf{1}_K^T \right); \mathbf{H}^* := \mathbf{M}^*; \mathbf{W}^* := \mathbf{M}^* \tag{5}$$

where $\mathbf{H}^*$ and $\mathbf{W}^*$ denote *target* state-action activations and *target* action selection layer weights, respectively.

## 4    MAIN APPROACH

### 4.1    ACTION COLLAPSE POLICY GRADIENT

The empirical experiments detailed in Sec.3 reveal that a gentle geometry, simplex ETF, emerges in ideal RL environments. Such a structure constitutes an optimal geometric structure for the action selection layer of optimal trained policy DNNs. This observation thus motivates the ACPG technique, where we explore affixing a synthetic ETF as our action selection layer and training our PG DNN from this initialization.

---

[4]The reward of Car-Racing is normalized to 0-20 for visualization.

[5]Because activations of Pong and Breakout is continuous and can be sampled in infinity times, mean of them are indeterminated value. We only show mean activations of Ideal Cliff Walking and Car-Racing.

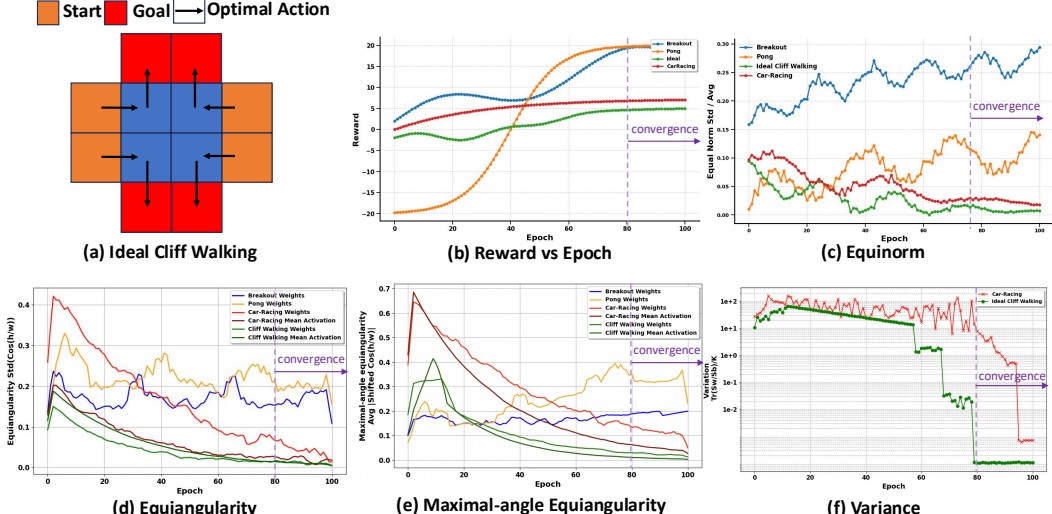

Figure 1: (a) Ideal environment diagram. (b) Models converge to stability; $y$-axis shows reward accumulation per epoch. (c) Weights of action selection layer become equinorm; $y$-axis shows $\text{Std}_k(\|w_k\|_2)/\text{Avg}_k(\|w_k\|_2)$. (d) Mean state-action activations and weights of action selection layer approach equiangularity; $y$-axis shows $\text{Std}_{k,k'\neq k}(\cos_{h/w}(k,k'))$. (e) Mean state-action activations and weights of action selection layer approach maximal-angle equiangularity; $y$-axis shows $\text{Avg}_{k,k'}|\cos_{h/w}(k,k')+1/(K-1)|$. (f) Variation of state-action activations sharing the same optimal action; $y$-axis shows $\text{Tr}\left\{\Sigma_W\Sigma_B^\dagger\right\}/K$, where $\text{Tr}\{\cdot\}$ is the trace operator, $\Sigma_W$ is the covariance of $h_k$, $\Sigma_B$ is the corresponding covariance between $h_k$, and $[\cdot]^\dagger$ is Moore-Penrose pseudo-inverse. In **Ideal Cliff Walking** and **Car-Racing**, Action Collapse happens in each figures. However, realistic RL environments such as **Breakout** and **Pong** show ever-decreasing equinormness and equiangularity which shows that they fall into biased optimal and do not form Action Collapse.

Specifically, we initialize the action selection layer $\mathbf{W}$ as a randomly oriented simplex ETF according to Eq.(3), scaled by a constant $\sqrt{E_\mathbf{W}}$ to enforce a fixed $\ell_2$ norm. During training, only state-action activations $\mathbf{H}$ are optimized. An overview of our method is illustrated in Fig.2. This selection layer outputs the likelihood of actions, from which an action is sampled and applied to the environment[6]. Although fixing the action selection layer as an ETF simplifies the learning problem, it brings theoretical merits and higher performance in our experiments.

## 4.2 THEORETICAL SUPPORT

In this subsection, we further explore the theoretical aspects of our framework, demonstrating that even in complex, real-world RL environments, Action Collapse is also achievable using a fixed ETF action selection layer. To facilitate tractable theoretical analysis, simplification is essential. The layer-peeled model (LPM), as proposed in Fang et al. (2021), offers an effective simplification by focusing solely on the action selection layer while disregarding the variable weights of the backbone layers. To capture key learning behaviors of the PG, we assume the DNN is modeled as an LPM and $S$ is defined in Sec.2. We allow $\mathcal{S}_k \subseteq \mathcal{S}$ and the possibility that the sizes of various $\mathcal{S}_k$ can be wildly imbalanced. Moreover, the action selection layer is fixed as a simplex ETF and only state-action activations $\mathbf{H} = \{h_s : s \in \mathcal{S}\}$ are optimized [7]. In this case, the $J(\mathbf{H})$ in Eq.(2) is represented as:

$$\max_{\mathbf{H}} \quad J(\mathbf{H}) = \sum_{s \in \mathcal{S}} d^\pi(s) \cdot \log\left(\frac{e^{h_s^T w_{a*}}}{\sum_{a \in \mathcal{A}} e^{h_s^T w_a}}\right) \Psi^\pi, \tag{6}$$

$$\text{s.t.} \quad ||h_s||^2 \leq E_\mathbf{H}, \ \forall k, k' \in [1, K], s \in \mathcal{S}, \tag{7}$$

---

[6]The agent's policy may not always select the highest-probability action. For instance, $\epsilon$-greedy exploration sometimes selects a random action to balance exploration and exploitation.

[7]$h_{s_k}$ is the state-action activation whose optimal action is $a_k$.

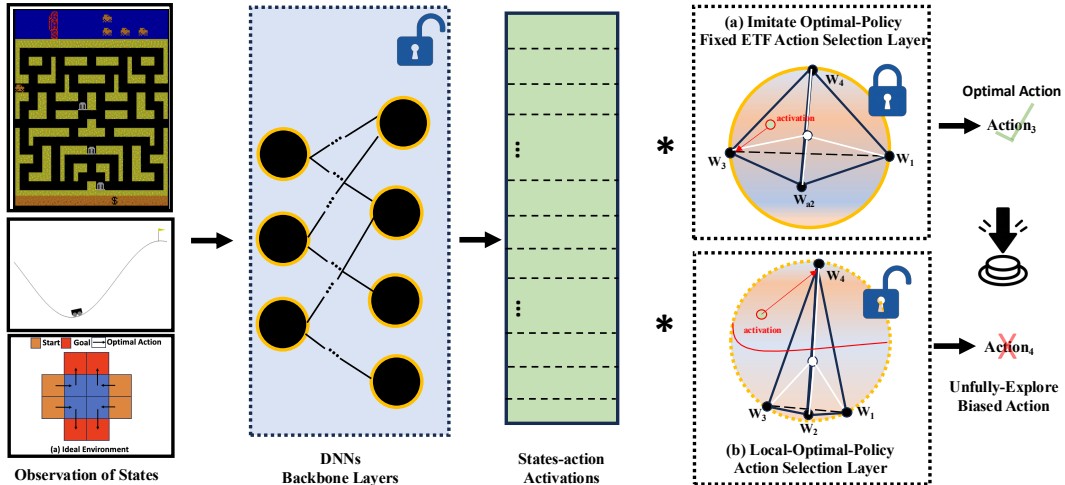

Figure 2: ACPG Framework. After observing states from the environment, the backbone layers can be trained to receive input states and the output hidden features aligned with any direction. action selection layer is a random simplex ETF and is **locked** during training. By contrast, normal DNNs employ a learnable action selection layer as shown in (b). Illustration of equiangular separation (a) and non-equiangular learnable separation (b). Action Collapse reveals (a), where the weights of action selection layer and mean state-action activations collapse to a simplex ETF. As some states are less explored than others in realistic environments, their weights lie in a closer to the origin, and the separability for these weights and mean activations degrades, as illustrated in (b). We can also see in (b), the green circle is biased toward $w_{a_4}$, which should instead perform $a_3$ as in (a).

where $a^*$ represents the optimal action of state $s$ and $d^\pi(s)$ represents the stationary distribution of the states. Note $\mathbf{W}^* = [w_1^*; \ldots; w_K^*]$ is the action selection layer as a simplex ETF with:

$$w_k^{*T} w_{k'}^* = E_{\mathbf{W}} \left( \frac{K}{K-1} \delta_{k,k'} - \frac{1}{K-1} \right), \quad \forall k, k' \in [1, K], \tag{8}$$

where $\delta_{k,k'}$ equals to 1 when $k = k'$ and 0 otherwise.

Despite the state set $\mathcal{S}$ being partially explored and the subset of states $\mathcal{S}_k$ corresponding to each optimal action $k$ being imbalanced, Eq.(6) still admits a global optimum, as shown in the following:

**Theorem 1.** *Whether or not 1) the states are fully explored; or 2) the action-state subsets $\mathcal{S}_k$ are mutually balanced; or 3) the action-state subsets $\mathcal{S}_k$ are fully-sized ($\mathcal{S}_k = \mathcal{S}$); any global minimizer $\mathbf{H}^* = [h_{s_k}^* : s_k \in \mathcal{S}]$ of Eq.(6) converges to a simplex ETF with the same direction as $\mathbf{W}^*$ with a length of $\sqrt{E_{\mathbf{H}}}$, i.e.,*

$$h_{s_k}^{*T} w_{k'}^* = \sqrt{E_{\mathbf{H}} E_{\mathbf{W}}} \left( \frac{K}{K-1} \delta_{k,k'} - \frac{1}{K-1} \right), \quad \forall \, 1 \le k, k' \le K, s \in \mathcal{S} \tag{9}$$

*where $s_k$ represents a state $s \in \mathcal{S}_k$. This indicates that Action Collapse emerges irrespective of the aforementioned conditions. This can be proven by converting to a conditional Lagrangian problem and applying Karush-Kuhn-Tucker (KKT) conditions. Please refer to Appendix C for our proof.*

Recalling the center computation in Eq.(1), we analyze the gradient of $J(\theta)$ w.r.t. the activation vector $h_s$ (recall this is short for $h(s; \theta)$) for each state $s$. Then the gradient of $J(\theta)$ has:

$$\nabla_\theta J(\theta) \propto \sum_{s \in \mathcal{S}} d^\pi(s) \mathbb{E}_\pi [\psi^\pi \nabla_\theta \ln \pi_\theta(a|s)] = \sum_{s \in S} d^\pi(s) \left( w_{k^*} - \frac{\sum_{j=1}^K w_j e^{h_s^T w_j}}{\sum_{j=1}^K e^{h_s^T w_j}} \right) \nabla_\theta h_s \psi^\pi$$

$$= \underbrace{\sum_{s \in \mathcal{S}} d^\pi(s) \left( \psi^+ w_{k^*} (1 - \pi_\theta(a_{k^*}|s)) \right) \nabla_\theta h_s}_{a_{k^*} \text{ is optimal for } s} - \underbrace{\sum_{s \in \mathcal{S}} d^\pi(s) \sum_{k=1, \neq k^*}^K \left( \psi^- w_k \pi_\theta(a_k|s) \right) \nabla_\theta h_s}_{a_k \text{ is not optimal for } s}$$

where $d^\pi(s)$ is stationary distribution of states. $\pi(a_k|s)$ is the policy probability of taking action $a_k$ at state $s$. As usual, Monte Carlo estimation is applied to approximate the formula inside the expected value. Note that $a_{k^*}$ indicates the optimal action for state $s$, which brings the highest return $\psi^+$, and other actions $a_k, k \neq k^*$ can bring lower return $\psi^-$.

It reveals that the gradient can be decomposed into two parts: the "optimal-action" part pulls gradient towards optimal action, $w_{a_k^*}$; while "taking other actions" part contains negative terms and pushes the gradient to the suboptimal action selection layer. This implies that once fully trained, the state-action activations will align with their corresponding action selection layer, forming a simplex ETF.

## 5 EXPERIMENTS

### 5.1 MAIN RESULTS

**Comparative Evaluation.** In Tab.1, we summarize the comparison results for two classic environments: discrete Cart Pole (Car.) and discrete Car-racing (Rac.), three Atari environments (Bellemare et al., 2013): EnduroNoFrameskip-v4 (End.), QbertNoFrameskip-v4 (Qbe.) and PongNoFrameskip (Pon.) with four popular PG algorithms: REINFORCE (Sutton et al., 1999), TRPO (Schulman et al., 2015), PPO (Schulman et al., 2017) and A3C (Mnih et al., 2016). To assess the generality and effectiveness of ACPG, two classes of Policy DNNs, multilayer perceptron (MLP) (Gardner & Dorling, 1998) and convolutional neural network (CNN) (Albawi et al., 2017) are employed. We applied ACPG to our existing DNNs while keeping the underlying policy-gradient algorithms unchanged. Across most CNN configurations, ACPG produced improvements of 5 – 15%. Notably, when combined with TRPO in the QBE., ACPG actually yielded a slight performance drop. We believe this is due to sampling bias: the highly imbalanced state–reward distribution makes it difficult to construct Action Collapse. From these experiments we observe:

1. Consistent performance gains: Across all four environments, ACPG maintains or increases its improvement.
2. Reduced variability: Most of the standard deviation of ACPG's results is uniformly lower than the baseline, indicating smoother and more reproducible convergence.

Higher reward and the reduced epochs showcase ACPG's effectiveness and convergence speed.

Table 1: Comparative analysis of ACPG performance against original algorithm (Org.) in gym. A detailed comparison of the best and final reward metrics achieved by MLP and CNN architectures, with and without ACPG integration. "Best" describes the highest reward during the whole training. "Final" is the reward at the last epoch. "Stop" denotes the early-stop epoch at which the reward has stabilized and the predefined performance target has been reached. "Std" describes the standard deviation of the final results over 20 runs with different seeds.

| Alg. | Env. | MLP | | | | | | | | CNN | | | | | | | |
|---|---|---|---|---|---|---|---|---|---|---|---|---|---|---|---|---|---|
| | | Best | | Final | | Stop | | Final Std | | Best | | Final | | Stop | | Final Std | |
| | | Org. | ACPG | Org. | ACPG | Org. | ACPG | Org. | ACPG | Org. | ACPG | Org. | ACPG | Org. | ACPG | Org. | ACPG |
| Rei. | Car. | 500 | 500 | 500 | 500 | 33 | 20 (43%+) | 2.47 | 1.54 | 500 | 500 | 500 | 500 | 65 | 15 (76%+) | 4.41 | 1.77 |
| | Rac. | 202.25 | 233.16 (15%+) | 158.72 | 196.15 (23%+) | 100 | 100 | 59.04 | 32.71 | 287.45 | 376.63 (31%+) | 188.94 | 284.73 (50%+) | 100 | 100 | 104.43 | 117.72 |
| PPO | End. | 941 | 1109 (17%+) | 884 | 1002 (13%+) | 500 | 500 | 143.23 | 96.41 | 1481 | 1509 (2%+) | 1483 | 1563 (5%+) | 100 | 100 | 15.43 | 20.28 |
| | Qbe. | 8685 | 10431 (20%+) | 6747 | 9450 (40%+) | 500 | 500 | 3584.31 | 1497.3 | 14852 | 15983 (7%+) | 14801 | 15779 (6%+) | 100 | 93 (7%+) | 103.37 | 91.40 |
| | Pon. | 10.9 | 14.7 (26%+) | 7.8 | 8.7 (12%+) | 500 | 500 | 4.95 | 4.21 | 20.1 | 21.1 (5%+) | 19.8 | 20.9 (5%+) | 100 | 89 (11%+) | 3.41 | 2.28 |
| TRPO | End. | 787 | 1290 (64%+) | 421 | 1257 (199%+) | 500 | 500 | 92.27 | 112.35 | 1243 | 1473 (19%+) | 1109 | 1309 (18%+) | 100 | 100 | 74.25 | 29.81 |
| | Qbe. | 13170 | 13007 (-0.01%) | 12954 | 8971 (31%-) | 500 | 500 | 4982.30 | 966.45 | 13787 | 14205 (3%+) | 13184 | 12589 (4%-) | 100 | 98 (2%+) | 166.52 | 107.35 |
| | Pon. | 10.4 | 13.5 (30%+) | 6.9 | 7.7 (12%+) | 500 | 500 | 5.32 | 3.37 | 20.1 | 20.9 (4%+) | 20.1 | 20.9 (4%+) | 99 | 89 (10%+) | 4.03 | 3.19 |
| A3C | Car. | 500 | 500 | 500 | 92 (8%+) | 100 | 100 | 0.23 | 0 | 500 | 500 | 500 | 500 | 100 | 43 (57%+) | 14.35 | 0 |
| | Rac. | 104.38 | 273.22 (161%+) | 113.54 | 190.28 (67%+) | 100 | 100 | 11.24 | 113.55 | 174.28 | 285.48 (64%+) | 191.52 | 241.65 (26%+) | 100 | 100 | 58.90 | 44.62 |

**Performance and Convergence.** Tab.1 demonstrates that ACPG markedly accelerates convergence, enhances robustness, and elevates cumulative rewards. In the more challenging Atari domains, ACPG outperforms standard policy-gradient baselines by margins of 199%, 64%, 19% and 18%. In the Car. environments, both ACPG and the original algorithm attain the maximal reward, reflecting these tasks' relative simplicity. But thanks to simplicity, we can closely analyze convergence differences in detailed reward-per-step time-series plots comparing ACPG with the original methods. In the Rac environment, ACPG achieves up to **161%** improvement over original A3C.

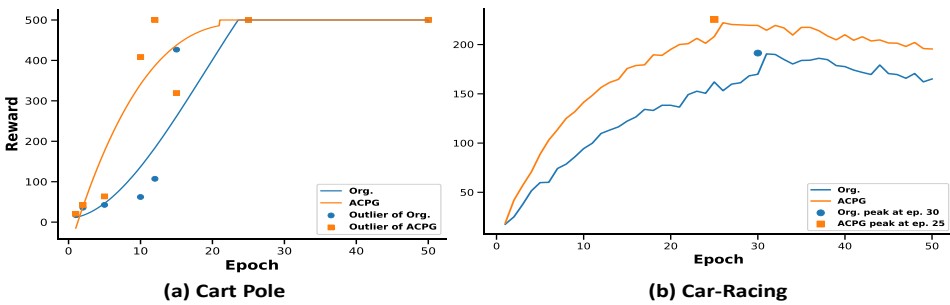

(a) Cart Pole        (b) Car-Racing

Figure 3: Convergence reward curve of Reinforce (Org.) and ACPG-augmented in the Cart Pole (Car.) and Car-Racing (Rac.) Environment. The result is averaged based on multiple experimental runs, each initialized with a distinct random seed. Outliers represent max/min reward achieved.

Fig.3 highlights ACPG's dramatic acceleration: by epoch 12, ACPG already reaches a nearly 12× improvement. Across key milestones, ACPG consistently delivers more than 8× higher rewards and converges in far fewer epochs. In the Car-Racing environment, ACPG similarly outpaces the baseline, achieving faster convergence and higher peak performance. ACPG's fixed ETF action selection layer precisely aligns activations with the Action Collapse targets. By contrast, conventional learnable layers frequently become trapped in local minima and require much longer training to disentangle state–action embeddings.

The remarkable improvement underscores ACPG's superior balance of exploration and exploitation: by actively probing under-explored states, it trains the action selection layer activations $h$ to better emulate the optimal policy. In comparison, baseline methods show pronounced performance volatility—especially in rare or typical states—because they often settle into local minima, which works well for familiar scenarios but leads to suboptimal action choices.

Table 2: Impact of exploration for ACPG in Pon. environment. $\epsilon$-greedy is integrated into PPO.

| Alg | Env. | Epochs | $\epsilon$ | CNN Best | CNN+ACPG Best |
|---|---|---|---|---|---|
| PPO | Pon. | 100 | 0.000 | 3.1 | 2.1 |
| | | | 0.001 | 7.1 | 7.2 |
| | | | 0.010 | 19.8 | 20.9 |
| | | | 0.100 | 18.5 | 21.6 |
| | | | 1.000 | 1.7 | 3.2 |

**Affect of Exploration Strategy.** To investigate exploration's impact on Action Collapse, we incorporated an $\epsilon$-greedy strategy (Sutton et al., 1999) into PPO—despite PPO's own softmax-based exploration. Tab.2 shows that, over 100 epochs, very low $\epsilon$ (i.e., minimal exploration) produces no performance gain when ACPG is applied. By contrast, as $\epsilon$ increases—especially around **0.001** — the ACPG yields significant improvements. We hypothesize that, because ACPG emulates the optimal policy, it demands adequate exploration to converge; but at $\epsilon = 1.0$, where sampling covers an excessively broad state distribution, both methods diverge. ACPG exhibits superior adaptability with higher Epsilon values, ensuring robustness and convergence amid a plethora of unseen states.

## 5.2 DOES ACTION COLLAPSE EXIST IN ACPG?

In Sec.3, Action Collapse does not naturally occur in Breakout and Pong. But under ACPG, the mean action selection layer activations trend towards equiangularity and maximal-angle equiangularity in Fig.4. While a fixed ETF action selection layer formally defines Action Collapse, it may not be the most effective configuration. In practice, settling at a saddle-point solution—rather than the global optimum—can yield more robust performance.

## 6 RELATED WORK

**Policy Gradient.** PG methods have garnered attention for their ability to scale gracefully to large spaces and incorporate deep networks as function approximators (Silver et al., 2014; Schulman et al.,

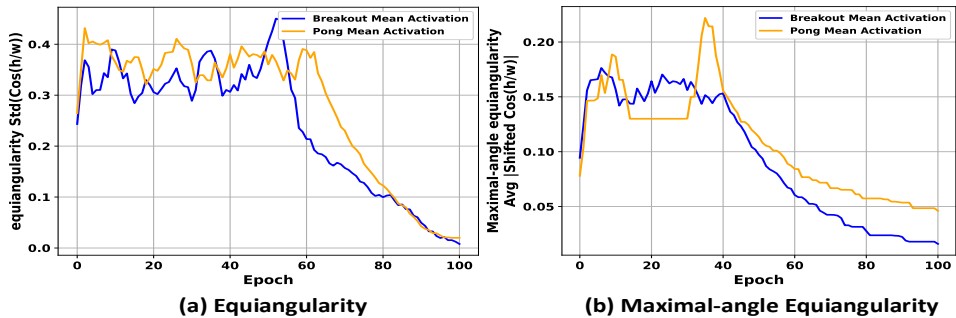

Figure 4: In Breakout and Pong, after applying ACPG, mean state-action activations approach equiangularity and maximal-angle equiangularity. Y-axis: $\text{Std}_{k,k'\neq k}(\cos_{h/w}(k,k'))$ and $\text{Avg}_{k,k'}|\cos_{h/w}(k,k')+1/(K-1)|$. Activations from training samples are averaged per epoch.

2015). REINFORCE (Sutton et al., 1999) relies on estimating returns through Monte-Carlo methods, while Actor-Critic methods (Konda & Tsitsiklis, 1999) approximate returns using another value network. Both of them are on-policy, with corresponding off-policy versions utilizing collected weighted trajectory rewards as returns. Asynchronous Advantage Actor-Critic (A3C) (Mnih et al., 2016) is an Actor-Critic method that generates sample batches by running multiple actors in parallel. DPG and DDPG (Silver et al., 2014; Lillicrap et al., 2015) model the policy as a deterministic decision, where the action is deterministically chosen. Schulman et al. (2015), Schulman et al. (2017) and Rafailov et al. (2023) proposed TRPO, PPO and DPO methods, which enforce a KL divergence constraint on policy update, helping avoid excessive parameter updates. While these previous PG algorithms study the effect of return function including update frequency, loss function and stability, they ignore the optimal geometric structure when applying DNNs.

**Neural Collapse.** NC is a phenomenon observed in the classification problem, which inspires us to explore neural network geometric structure in RL. Papyan et al. (2020) first discovered the NC phenomenon. At the TPT, a classification model trained on a balanced dataset exhibits the collapse of last-layer features into their within-class centers. These centers and classifiers will form a simplex ETF. Due to its elegant geometric property, recent studies prove it with the assumption of last-layer features and the last-layer as independent variables under CE loss (Fang et al., 2021; Lu & Steinerberger, 2020) and MSE loss (Mixon et al., 2020; Han et al., 2021). Some studies try to induce neural collapse in imbalanced learning for better accuracy of minority classes (Liu et al., 2023; Liang & Davis, 2023; Yang et al., 2022). Besides, NC has also been observed during LLM training process, which is linked to increasing generalization (Wu & Papyan, 2024). By contrast, we discover the structures of the last-layer feature means and action selection layer in PG and propose ACPG, which allows the policy to imitate the optimal policy.

Other related works - **Optimality and Convergence Analysis**, **Exploration and Exploitation** and **Reinforcement Learning in Large Language Model** are is included in Appendix E

## 7    CONCLUSION

In this paper, we explore the Action Collapse for RL PG methods. To the best of our knowledge, this is the first work to investigate the geometric properties of optimal policy DNNs. We begin by empirically observing the Action Collapse phenomenon and subsequently extend our analysis in complex discrete RL environment. In complex settings, where $\mathcal{S}$ is not fully explored and certain conditions in empirical experiments are not satisfied, the Action Collapse is disrupted. To approach Action Collapse, we introduce a synthetic fixed simplex ETF as action selection layer. We conclude that our simplified practice even helps to improve the performance of PG methods with no cost. We anticipate that our findings will inspire future theoretical investigations into RL's optimal policy. The limitation of this study is its analysis of Action Collapse predominantly conducted under certain conditions. Future research should aim to investigate more intricate RL DNNs, more complex and continuous environments, and PG loss that are more conducive to facilitating Action Collapse.

## 8    ETHICS STATEMENT

We have read and will adhere to the ICLR Code of Ethics. This work does not involve human participants, personally identifiable information, or any user-generated content. All data used are standard, publicly available benchmarks obtained under their respective licenses; no demographic attributes are collected or inferred. Our methods focus on model architecture/optimization and do not introduce capabilities aimed at surveillance, profiling, or other harmful applications. We assess no foreseeable risks related to privacy, security, discrimination/fairness, or legal/regulatory compliance in the scope of this study, and no IRB/ethics approval was required. We will release code, configuration files, and detailed instructions to support transparency and reproducibility, and we certify that results are reported honestly without fabrication or inappropriate manipulation. We have no conflicts of interest or external sponsorship that could bias the work.

## 9    REPRODUCIBILITY STATEMENT

We provide an anonymized repository (URL in the supplementary material) with full training and evaluation code for *MLP, DQN* methods across *Rei, PPO, TROP, A3C*. Our code includes exact configuration files in Tables 1/Figures [F1–F4], scripts to download and verify datasets, deterministic preprocessing, fixed random seeds, and environment specifications (python requirements.txt). Algorithmic details appear in §[3]; dataset descriptions, licenses, and splits are given in §[4] and Appx. [B]; hyperparameters and search ranges are listed in Appx. [A]. Running *python NeuralCollapse4RL/python_parallel_run.py* recreates reported metrics and regenerates plots and logs (including seeds and software versions). We report means and 95% confidence intervals over *20* runs; hardware and runtime details are in Appx. [C]. Any deviations from defaults are noted in the README. Upon acceptance, we will release a de-anonymized public repository under the *Apache license 2.0*.

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

# Imitate Optimal Policy: Prevail and Induce Action Collapse in Policy Gradient

## *Supplementary Material*

## A  SCOPE CLAIRFY

Our paper makes a single, precise claim: we give (to our knowledge) the first rigorous proof of Action Collapse under the policy-gradient loss in discrete MDPs, and we translate that insight into the ACPG algorithm that enforces the predicted simplex-ETF geometry at the policy head. This "prove the geometry $\rightarrow$ bake it into the head" recipe yields faster, stabler learning in the setting our theory covers.

We want to clarify our scope: our whole strict proof is based on discrete environments. Tons of fundamental RL algorithms arise from a discrete environment setting. We have revised the paper to clearly state the assumptions (discrete actions, PG loss) up front and we want to state continuous settings are out of scope for our theorem. We also added a new discussion section that sketches a possible optimal structure to a continuous action environment and lays out a roadmap for extending our proof-then-fix methodology beyond policy gradient loss in App. F.

Our contribution is a theoretical phenomenon + geometry-aware head for discrete PG, analogous in spirit to how Neural Collapse first crystallized on idealized settings before inspiring broader practice. We believe the community benefits from isolating and proving such structures cleanly, then expanding them in subsequent work.

## B  NEURAL COLLAPSE PHENOMENON

Then the neural collapse (NC) phenomenon can be formally described as (Papyan et al., 2020):

**(NC1)** Collapse of within-class variability: $\Sigma_{\mathbf{W}} \rightarrow \mathbf{0}$, and $\Sigma_{\mathbf{W}} := \mathrm{Avg}_{i,k}\{(h_{k,i} - h_k)(h_{k,i} - h_k)^T\}$, where $h_{k,i}$ is the last-layer feature of the $i$-th sample in the $k$-th class, and $h_k = \mathrm{Avg}_i\{h_{k,i}\}$ is the within-class mean of the last-layer features in the $k$-th class;

**(NC2)** Convergence to a simplex ETF: $\tilde{h}_k = (h_k - h_G)/||h_k - h_G||, k \in [1, K]$, satisfies Eq. (4), where $h_G$ is the global mean of the last-layer features, *i.e.,* $h_G = \mathrm{Avg}_{i,k}\{h_{k,i}\}$;

**(NC3)** Self duality: $\tilde{h}_k = w_k/||w_k||$, where $w_k$ is the classifier vector of the $k$-th class;

**(NC4)** Simplification to the nearest class center prediction: $argmax_k\langle h, w_k\rangle = argmin_k||h - h_k||$, where $h$ is the last-layer feature of a sample to predict for classification.

## C  PROOF OF THEOREM 1

We consider the optimization problem:

$$\max_{\mathbf{H}} J(\mathbf{H}) \tag{10}$$

where $\mathbf{H} = \{h_{s_k} : s \in \mathcal{S}\}$ represents the activation of states with optimal action $a_k$. Expanding the policy gradient reward function:

$$\max_{\mathbf{H}} \sum_{k=1}^{K} \sum_{s_k} d^\pi(s_k) \cdot \log \frac{e^{h_{s_k}^T w_k}}{\sum_{j=1}^{K} e^{h_{s_k}^T w_j}} \Psi^\pi \tag{11}$$

Rewriting in terms of state-actionactivations distributions, where we separate the states $s$ into their corresponding optimal action $a_k$:

$$\max_{\mathbf{H}} \sum_{k=1}^{K} \sum_{i=1}^{n_k} d^{\pi}(s) \cdot \log \frac{e^{h_{i,k}^T w_k}}{\sum_{j=1}^{K} e^{h_{i,k}^T w_j}} \Psi^{\pi} \tag{12}$$

where $h_{i,k}$ represents the state-action activations of the $i^{th}$ state in class $\mathcal{S}_k$, and $n_k = |\mathcal{S}_k|$ represents the cardinality (size) of class $\mathcal{S}_k$. Rearranging (by dividing $e^{h_{i,k}^T w_k}$ from both numerator and denominator):

$$\max_{\mathbf{H}} \sum_{k=1}^{K} \sum_{i=1}^{n_k} d^{\pi}(s) \cdot \log \frac{1}{1 + \sum_{j \neq k} e^{h_{i,k}^T (w_j - w_k)}} \Psi^{\pi} \tag{13}$$

which is equivalent to minimizing:

$$\min_{\mathbf{H}} \sum_{j \neq k} e^{h_{i,k}^T (w_j - w_k)} \tag{14}$$

where we only focus on optimizing the value of $\mathbf{H} = \{h_{i,k}\}$. Thus, we only maintain the $i^{th}$ sample in class $k$, and write $h_{i,k}$ as $h$ for simplicity. Note that for continuous state space, this formula also holds, since we use integral instead of sum to overwrite $J(\mathbf{H})$. Then we rewrite to:

$$\min_{h} \sum_{j \neq k} e^{h^T (w_j - w_k)} \tag{15}$$

Define the function:

$$f(h) = \sum_{j \neq k} e^{h^T (w_j - w_k)} \tag{16}$$

subject to the constraint:

$$g(h) = \|h\|^2 - E_{\mathbf{H}} \leq 0 \tag{17}$$

Construct the Lagrangian:

$$\mathcal{L}(h, \lambda) = f(h) + \lambda g(h) \tag{18}$$

Using Karush-Kuhn-Tucker (KKT) conditions, we require:

$$\frac{\partial}{\partial h} \mathcal{L}(h, \lambda) = \sum_{j \neq k} (w_j - w_k) e^{h^T (w_j - w_k)} + 2\lambda h = 0 \tag{19}$$

The KKT conditions require:

$$\lambda g(h) = 0, \quad g(h) \leq 0, \quad \lambda \geq 0 \tag{20}$$

Note both $f(h)$ and $g(h)$ are convex functions, since they both have non-negative second derivatives (which can be easily verified by the second derivative of $(e^{h^T (w_j - w_k)})'' = (w_j - w_k)^2 e^{h^T (w_j - w_k)} \geq 0$ in $f(h)$ and $(h^2)'' = 2 > 0$ in $g(h)$). Since KKT conditions are both necessary and sufficient for convex problems , we proceed with:

**Lemma 1.** $\sum_{k=1}^{K} w_k = 0$.

*Proof.* The Gram matrix for a set of simplex ETF:

$$G_{ij} = \begin{cases} E_{\mathbf{W}}, & i = j \\ -\frac{E_{\mathbf{W}}}{K-1}, & i \neq j \end{cases} \tag{21}$$

It can be verified that Gram matrix $G$ has all-ones vector $\mathbf{1}$ as an 0-eigenvector since:

$$G\mathbf{1} = E_{\mathbf{W}} + (K-1)(-\frac{E_{\mathbf{W}}}{K-1}) = 0 \tag{22}$$

Substituting $G = \mathbf{W}^T\mathbf{W}$ following the definition of simplex ETF we have:

$$G\mathbf{1} = 0 \Rightarrow \mathbf{W}^T\mathbf{W}\mathbf{1} = 0 \tag{23}$$

This can be simplified from how we construct the ETF matrix $\mathbf{W}$. We assume $\mathbf{W} \in R^{d \times K}$, and for any activation dimension $d > K - 1$, we add 0 to the row greater than $K - 1$ and we keep the elements in first $K - 1$ rows in each column to be the tightest simplex ETF (i.e. $K$ simplex ETF vectors in a $K - 1$ space). Therefore, proving the sum of action selection layer to be 0-vector needs only to be considered in the case where $\mathbf{W} \in R^{(K-1) \times K}$ is in the tightest shape, since the other components with row index greater than $K - 1$ are set to 0.

Therefore, to solve the equation $\mathbf{W}^T a = 0$ for some vector $a \in R^{K-1}$, we have: $a = 0$ since the vector $\mathbf{W}^T$ have full rank in its column, indicating it has only 0-vector in its null space. This indicates that:

$$\mathbf{W}^T\mathbf{W}\mathbf{1} = 0 \Rightarrow \mathbf{W}\mathbf{1} = 0 \tag{24}$$

The above equation simplifies to $\sum_{j=1}^{K} w_j = 0$.

$\square$

**Proposition 1.** *KKT holds based on the condition:*

$$h = \sqrt{\frac{E_{\mathbf{H}}}{E_{\mathbf{W}}}} w_k, \tag{25}$$

*Proof.*

$$\sum_{j \neq k} (w_j - w_k) e^{w^T(w_j - w_k)} + 2\lambda h = 0 \tag{26}$$

We define:

$$A = e^{\sqrt{\frac{E_{\mathbf{H}}}{E_{\mathbf{W}}}} w_k^T(w_j - w_k)} \tag{27}$$

where A is a constant independent of $j$ since $w_k^T w_{k'} = -\frac{E_{\mathbf{W}}}{K-1}$ if $k \neq k'$ and $E_{\mathbf{W}}$ otherwise. And we rewrite the formula as:

$$A \sum_{j \neq k} (w_j - w_k) + 2\lambda h = 0. \tag{28}$$

Rearranging and substituting $\sum_{j=1}^{K} w_j = 0$ from the Lemma and condition $h = \sqrt{\frac{E_{\mathbf{H}}}{E_{\mathbf{W}}}} w_k$:

$$2\lambda \sqrt{\frac{E_{\mathbf{H}}}{E_{\mathbf{W}}}} w_k = AKw_k, \tag{29}$$

where we pick the optimal $\lambda$ to be:

$$\lambda = \frac{1}{2} \sqrt{\frac{E_{\mathbf{W}}}{E_{\mathbf{H}}}} AK > 0. \tag{30}$$

The norm constraint is active, indicating $g(h) \leq 0$ since:

$$g(h) = \|h\|^2 - E_{\mathbf{H}} = \frac{E_{\mathbf{H}}}{E_{\mathbf{W}}}\|w_k\|^2 - E_{\mathbf{W}} = 0, \tag{31}$$

also $\lambda = \frac{1}{2}\sqrt{\frac{E_{\mathbf{W}}}{E_{\mathbf{H}}}}AK > 0$, all KKT conditions are satisfied, proving:

$$h = \sqrt{\frac{E_{\mathbf{H}}}{E_{\mathbf{W}}}}w_k. \tag{32}$$

is the optimal solution (from our previous argument, in convex case, satisfying KKT condition is necessary and sufficient to show optimal solution of $h$ and $\lambda$).

$\square$

The proposition above satisfies the condition we aim to prove in the theorem:

$$h^T w_{k'} = \sqrt{E_{\mathbf{H}}E_{\mathbf{W}}}\left(\frac{K}{K-1}\delta_{k,k'} - \frac{1}{K-1}\right). \tag{33}$$

where $h = h_{i,k}$ or $h_{s_k}$ represents the activation of a state in $\mathcal{S}_k$

$\square$

Note: Notable differences exist between the loss function in Eq.(1) and Eq.(2). The key difference is, Eq.(1) follows the state value function as the loss function, which includes all the action choices in the formula. By contrast, Eq.(2) leverages PG, which solely focusing on optimizing the policy probability. Both formulations can serve as valid loss functions for policy training. In fact, the PG formulation is derived from the value function using Monte Carlo estimation, where the optimal action choice acts as an unbiased estimator for the expected value. In this paper, PG is used as the loss function in reaching the Theorem, and value function is expanded to show the phenomenon that activations will be attracted to align with the simplex ETF structure.

## D    EXPERIMENTS DETAILS

All experiments were conducted on servers equipped with NVIDIA H100 80 GB GPUs paired with dual Intel Xeon Platinum 8462Y+ processors (2 × 32-core sockets, 64 cores total) and approximately 2 TB of RAM.

All hyper-parameters are shown as below:

- The MLP Classic Control / box2d: MLP hidden sizes: [64, 64], Batch size: 64, Buffer size: 20000, Activation: ReLU, Output: Softmax for policy.
- The MLP Atari: Hidden sizes: [N, 256, 128, 256, N] (N is based on the feature dimension after CNN process image), Activation: ReLU, Output: Linear for Q-values.
- The CNN: Conv layers: [256->128->32->64->64] channels, Kernel sizes: [8->4->3], Strides: [4->2->1], Hidden size: [512], Then following MLP above.
- Ideal Cliff Walking: Epochs: 100/500, Step per epoch: 1000, Reward threshold: 195.
- Car-racing: Epochs: 100, Step per epoch: 100, Step per collect: 10, Repeat per collect: 2, Training num: 20, Test num: 20. **We constrained setting: each episode is limited to 100 training steps (vs. the usual 1000 frames + 10000 steps * 10 epochs)**
- Break-out: Epochs: 100/500, Step per epoch: 100000, Step per collect: 1000, Repeat per collect: 4, Training num: 20, Test num: 20.
- End.: Epochs: 100/500, Step per epoch: 100000, Step per collect: 1000, Repeat per collect: 4, Training num: 20, Test num: 20
- Qbe.: Epochs: 100/500, Step per epoch: 100000, Step per collect: 1000, Repeat per collect: 4, Training num: 20, Test num: 20

- Pon.: Epochs: 100/500, Step per epoch: 100000, Step per collect: 1000, Repeat per collect: 4, Training num: 20, Test num: 20
- Reinforce hyper-parameters: Learning rate: (1e-3, 1e-4, 1e-5), Gamma: 0.95, Batch size: 64, Buffer size: 20000, Training episodes per collect: 8, Repeat per collect: 2
- PPO hyper-parameters: Learning rate: (8e-5, 2.5e-4, 8e-4) Gamma: 0.99, GAE lambda: 0.95, Clip epsilon: 0.1, Value function coefficient: 0.25, Entropy coefficient: 0.01, Max gradient norm: 0.5, Batch size: 256, Buffer size: 100000, Training episodes per collect: 10, Repeat per collect: 4
- TRPO hyper-parameters: Learning rate: (8e-5, 2.5e-4, 8e-4),Gamma: 0.99,Max KL divergence: 0.01, Value function coefficient: 0.5, Entropy coefficient: 0.01, Batch size: 256, Buffer size: 100000, Training episodes per collect: 10, Repeat per collect: 4
- Batch size: 16 (classic control) / 128 (Atari) over all environments.

All experimental results are average over **20** random seeds and we choose the best from three learning rates. And our anonymized repository: LINK

## E  RELATED WORK

**Optimality and Convergence Analysis** Policy training requires the action distribution to converge either to the global optima or a stationary point (Xiong et al., 2022b). However, understanding of the optimal geometric structure is limited. Theorem 4.2 of Zhang et al. (2019) establishes the asymptotic convergence of random-horizon PG algorithm. Additionally, Zhang et al. (2020) proposes *Modified Random-Horizon Policy Gradient* (MRPG) algorithm to approximately converge to a second-order stationary point. To ensure convergence, objective functions such as regularizing entropy are widely accepted and proven to be less likely converge to local optima, offering a robust optimization (Ahmed et al., 2019; Husain et al., 2021). The deterministic policy can be optimized with a guaranteed stationary solution (Xiong et al., 2022a), but this usually converges in a local optimum in practice. When there is a gradient dominance condition, Bhandari & Russo (2019; 2021) studied linear convergence rate for PG methods. Nevertheless, all these works only study how and when the agent model can converge to the stationary point or global optimal, but ignore the optimal geometric structure of model in TPT.

**Exploration and Exploitation** Several theoretical analyses in reinforcement learning aim to comprehend how to explore and exploit environmental data for improving model training. Optimism in the Face of Uncertainty (OFU) is a method based on confidence (Strehl & Littman, 2005; Simchowitz & Jamieson, 2019; Dann & Brunskill, 2015) or bonus (Strehl et al., 2006; Jin et al., 2018a). While they have strong theoretical analysis and guarantees, applying them in deep RL presents challenges due to the need for preserving visits for each state in a tabular way. Specifically to PG, theoretical analysis of entropy maximization indicates increasing action entropy only promotes undirected exploration. Therefore, it is suggested to balance the random choice and the output of policy model to encourage more exploration while maintaining exploiting existing network.

**Reinforcement Learning in Large Language Model** RL, especially RLHF, has been widely applied to train LLM, where RL favors the process of improving the efficiency of human feedback. Human values and preferences are used in improving the reward function accuracy, where a latent variable framework can be introduced for better encoding (Sriyash et al., 2024). Even the need of explicit reward function can be replaced by preference data using a classification loss, with the help of *direct preference optimization* (DPO) (Rafailov et al., 2023). Also, safety issue in LLM training and development is currently more and more important, which can be formalized and included in reward function to improve helpfulness and harmlessness, utilizing existing methods in RL to favor the optimization (Dai et al., 2024). Harnessing the advantage of PG, methods such as *contrastive policy gradient* (CoPG) are applied in fine-tuning and output human preferences more effectively (Flet-Berliac et al., 2024) . These examples demonstrate the importance of RL in LLM training, which indicates the potential application of our ACPG method.

## F  LIMITATION AND SOCIETAL IMPACTS

The potential limitations of our study may include:

- Our supposition of Action Collapse has only been shown to happen in ideal discrete environments, while others have not yet been tested. We conjecture that if the environment becomes more complex, with more random transition and sparse rewards, this phenomenon may be difficult to observe or even does not exist. Furthermore, what does geometric structure characterize at a saddle (sub-optimal) point?

- Theorem 1 1 states our goal is optimized when last layer activations match with their preassigned action selection layer, but have not yet been tested whether arbitrary states from the same optimal-action class will finally collapse to corresponding vector in TPT.

- Although experiments have demonstrated better adaptability in Sec.5.1 by comparing PPO with fixed action selection layer or not, other PG methods has not yet been fully tested by applying fixed action selection layer. The impact of construction of sampling RL dataset also warrants further investigation.

- Explore optimal structure under MSE loss and continuous environments: Similarly to our simple environment, rather than hard-coding the frame structure, an alternative is to learn or discover the optimal geometric structure of actions by optimizing a suitable loss (such as mean squared error) in a continuous environment. Similarly to our simple discrete setting, where the best performing policies utilized actions arranged in an equiangular configuration (minimizing overlap and redundancy), we expect that in a continuous domain, the optimal arrangement of policy outputs (in a mean-square sense) is an isotropic, low-coherence distribution of actions. In other words, under an MSE objective, the policy should ideally produce a set of basis actions that are as orthogonal or as evenly separated as possible across the action manifold. This intuition is supported by frame theory: minimizing pairwise coherence is known to reduce error in signal representation tasks *On the structures of Grassmannian frames John I. Haas IV, Peter G. Casazza*

- Our initial theoretical observations, such as fully explored state spaces and uniform initial state distributions, are limited in idealized discrete conditions. Nonetheless, these conditions allow us to precisely identify and analyze the phenomenon. Extending these insights into more complex, real-world scenarios—such as robotics—would significantly increase practical applicability.

**Societal Impacts** Our work introduces Action Collapse Policy Gradient (ACPG), a broadly applicable paradigm for training discrete-policy RL networks by fixing the action selection layer to a simplex ETF structure. This approach offers theoretical insights into optimal policy geometry and yields practical benefits—namely faster convergence and more stable learning across a wide range of Gym environments—without altering the core interaction loop or data requirements of existing PG methods. Because ACPG only replaces the action selection layer with a non-learnable ETF, it often lowers overall computation and energy usage compared with conventional learnable action selection layer, imposing no extra environmental burden. Beyond these efficiency gains, our method does not introduce new concerns in areas such as privacy, public health, fairness, or other societal domains: it remains fully compatible with standard RL safety and ethics frameworks and does not affect the downstream interpretability or deployment of learned policies.

## G  LARGE LANGUAGE MODELS USAGE

We used a large language model - ChatGPT (GPT-5 thinking) solely for grammar and spelling edits to author-written text. We used Claude Code to assist code writing. The tool did not generate scientific content, design experiments, analyze data, or select citations, and therefore did not contribute at the level of a contributing author. All edits were reviewed and approved by the authors, who take full responsibility for the final manuscript.

