# OpenReview forum: "Imitate Optimal Policy: Prevail and Induce Action Collapse in Policy Gradient"
_ICLR.cc/2026/Conference — Submitted to ICLR 2026_

### Official Review · Reviewer_pMkN · 2025-10-28

**Soundness:** 1
**Presentation:** 2
**Contribution:** 2
**Rating:** 2
**Confidence:** 3

**Summary:**

In some toy examples under hard constraints, the author observed Action Collapse where DNN acts as a reinforcement learning policy.
Based on this observation and theory in Neural Collapse, the author proposes a method which fixes the structure of action selection layer.
Experiments are done in the discrete case, containing two Gym classic envs and 3 Atari envs.

**Strengths:**

1.This paper introduce the concept of neural collapse into Deep Reinforcement Learning.
2.This paper reveal that a simplex equiangular tight frame (ETF) will appear in the action selection layer under some constraints, which is similar as neural collapse.

**Weaknesses:**

1. **Why does "using the ETF structure as a fixed configuration in the action selection layer" work?**
   The author dedicates a section to demonstrating that Action Collapse will occur in the reinforcement learning policy network, but this doesn’t fully explain why the approach works. Could you clarify the underlying mechanism more explicitly?

2. **Why was the policy gradient RL method chosen to integrate Action Collapse?**
   Since the Action Collapse is primarily aimed at optimizing classification DNNs, why not use Deep Q-Networks (DQN) as the first approach in RL? DQN seems like a more suitable method to integrate with Action Collapse. Could you elaborate on the reasoning for this choice?

3. **What is the distinction between your method and existing research on neural collapse?**
   Is your work essentially integrating the existing method (fixed ETF in the last layer) into reinforcement learning? It would be helpful to highlight the novel contributions of your work compared to prior research in this area.

4. **What are the benefits of using a fixed ETF in the last layer?**
   Similar to Question 1, could you provide some intuitive analysis or experimental results to illustrate why this approach is effective? Understanding the rationale behind using the fixed ETF would strengthen the argument for its benefits.

5. **Can this method be applied to continuous action spaces?**
   It seems that the ETF structure is mainly applicable in discrete action spaces. Can you clarify if and how this approach can be extended to continuous action spaces?

6. **The experimental domains are relatively simple.**
   The experiments are conducted in overly simple settings. More complex experiments are needed to demonstrate the robustness and performance of the proposed method.
It seems the method only requires a modification of the network structure, which will not bring a heavy burden for the author.

7. **If the proposed method doesn't perform well in more complex experiments, what are the underlying reasons?**
   In such cases, an ablation study and a broader range of experiments would be valuable to pinpoint the specific scenarios where the method succeeds or fails. This could provide insightful conclusions for the RL community and help understand the method’s limitations better.


Another kind Tips:
Try not use "Our early work"(line 021 Abstract) in double-blind review process

**Questions:**

see above

---

### Official Review · Reviewer_MtCv · 2025-10-29

**Soundness:** 1
**Presentation:** 2
**Contribution:** 1
**Rating:** 2
**Confidence:** 4

**Summary:**

The author introduce neural collapse, a geometric property of the final activation and weights of a fully trained classification NN, and propose a similar idea, action collapse, in policy gradient policies.  They conduct empirical analysis in a set of toy tasks with observe action collapse and propose a method to induce action collapse in policy gradient methods by setting the final weight layer to an ETF and freezing it, thereby enforcing one of the action collapse properties.  They provide theoretical results to show that other properties are fulfilled and provide empirical results in discrete control and atari tasks that their method improves upon the base policy gradient method.

**Strengths:**

- Authors propose a simple, computationally cheap method that's compatible with existing policy gradient methods.
- Promising experimental results over multiple algorithms and tasks.

**Weaknesses:**

1. The proposed Action Collapse is not well-defined.  It's never formally defined in the paper and the distinction to noise collapse is not clear.  Furthermore in Section 3, it's unclear how the authors are interpreting their results and drawing the conclusion that action collapse is present in RL policies because action collapse is never defined.

2. The policy gradient loss function the authors define in Equation 2, and build off of in Section 4.2 to derive theoretical results is wrong.  The $$pi_\theta(a|s)$$ term should come from the actions taken by the policy, not the optimal action $$pi_\theta(a^*|s)$$.  (In fact, we do not know $$a^*$$ in RL).  This invalidates their argument that policy gradient is Q-value weighted classification loss and the results in section 4.2

3. Enforcing action collapse in RL policies is not well-motivated.  The presence of action collapse in fully converged policies does not necessarily mean this property is desirable during training.  In fact, RL often uses an entropy regularization loss to incentivize exploration during training.  The proposed method of freezing the final weight layer would also reduce model capacity.

4. The experiments are fairly limited and should include more tasks and evaluate each method over multiple tasks.  Furthermore the scope of the method may be limited to only discrete action spaces and should be clearly stated.

**Questions:**

1. Does this method apply only to policy gradient methods for discrete action spaces?  Could it be combined with SAC?
2. Are there learning curves for Table 1?  How were the hyperparameters for each method tuned?  I'm curious how ACPG has 200% performance in some cases with a smaller capacity network.
3. In Section 3, why not just check if the final weight layer is a simplex ETF?

---

### Official Review · Reviewer_398U · 2025-11-01

**Soundness:** 4
**Presentation:** 3
**Contribution:** 3
**Rating:** 8
**Confidence:** 3

**Summary:**

The authors identify a new phenomenon for RL training, termed "Action Collapse" (AC), which is analogous to "Neural Collapse" (NC) seen in classification tasks. In ideal training scenarios, they observe that the final-layer representations of an optimal policy network exhibit AC, characterized by equinorm and equiangularity, and converge to form a simplex equiangular tight frame (ETF).

By constraining the weight in the final action-selection layer to be an ETF, the newly proposed algorithm, ACPG, which integrates three PG algorithms, shows consistent performance gains in two classic control environments and two Atari environments under both MLP and CNN frameworks.

**Strengths:**

The paper clearly explains the action collapse (AC) phenomenon, with a clear demonstration in ideal RL environments.

Their algorithm is well-motivated by empirical observations and is well-supported theoretically; the ACPG method induces Action Collapse, as proven in Theorem 1.

The emergence of the AC phenomenon with ACPG on Pong and Breakout empirically supports their theorem.

**Weaknesses:**

1. The paper's core theoretical contribution, the identification of "Action Collapse" (AC) and the proof for Action Collapse Policy Gradient (ACPG) (Theorem 1), is strictly limited to discrete action spaces.

2. Unclear Reliance on Full Exploration: As shown in Table 2, ACPG works under sufficient exploration. This leads to an issue with extremely large action spaces.

**Questions:**

Figure 1(e) is titled "Maximal-angle Equiangularity" with a mathematical equation provided. Can you explain how this equation represents the max-angle?

---

### Official Review · Reviewer_izdB · 2025-11-01

**Soundness:** 2
**Presentation:** 2
**Contribution:** 2
**Rating:** 2
**Confidence:** 4

**Summary:**

The paper introduces the Action Collapse Policy Gradient (ACPG) method, which modifies the action selection layer of a policy gradient model by constraining it to a synthetic simplex ETF (Equiangular Tight Frame). The authors present a theoretical analysis suggesting that ACPG encourages activation means to collapse to a simplex ETF structure. Experimental results on discrete reinforcement learning (RL) benchmarks are reported, showing higher returns and improved convergence speed compared to prior policy gradient methods.

**Strengths:**

- Addresses the challenge of improving stability and convergence in policy gradient methods.
- Seeks to connect representation geometry with reinforcement learning optimization.
- Provides both theoretical discussion and empirical evaluation on discrete environments.

**Weaknesses:**

- Central concepts and definitions require clearer formalization and interpretation.
- The generality of the proposed approach beyond discrete settings is not established.
- The empirical evidence does not fully support several of the stated claims.
- The relation between the theoretical framework and its practical implementation is not sufficiently explained.
- The presentation of results could be expanded to better illustrate the method’s behavior and evaluation process.

**Questions:**

**Detailed Review:**

The paper proposes ACPG, a policy gradient variant that constrains the action selection layer to follow a synthetic simplex ETF structure. The theoretical section suggests that mean activations collapse toward an ETF configuration, implying that “action collapse” emerges naturally during training. However, the notion of action collapse is not precisely defined, and the connection between this geometric concept and standard policy optimization objectives is not clearly developed.

In the experimental section, results are presented for discrete environments such as Breakout and Pong. The paper claims that these are “realistic environments” but does not explain what properties make them so, or how equinormness and equiangularity are computed. The reasoning behind Figure 1 and its message should be clarified, as it is not obvious what behavior or phenomenon it illustrates.
Certain claims, such as the statement that baseline methods exhibit volatility due to settling into local minima, are made without sufficient evidence or quantitative justification. It is unclear whether ACPG indeed achieves optimal results or whether observed differences arise from implementation or tuning choices. Moreover, as the reported scores are not normalized, performance comparison across tasks remains ambiguous.

The discussion of RL applications to LLM training appears peripheral and should be either removed or better justified within the paper’s main context. If the goal is to show ACPG’s generality, a clearer connection to the underlying reinforcement learning principles should be established.

Finally, the description of the ETF constraint integration into policy gradient methods such as PPO is incomplete. Details on how simplex ETF parameters are computed and incorporated into the optimization are critical for reproducibility. Similarly, Table 1 presents only final scores, omitting learning curves that could illustrate training dynamics. The stopping criterion (“Stop”) is also unclear, and it is not specified whether it follows a systematic approach or an ad hoc rule.

**Questions:**
1. Could the authors clarify how action collapse is formally defined and measured during training?
2. How can ACPG be extended or adapted to continuous action spaces?
3. What properties make environments like Breakout and Pong “realistic,” and how do these relate to the observed behavior of action collapse?
4. Could the authors elaborate on the interpretation of Figure 1 and the insight it is intended to convey?
5. How are the ETF constraints computed and integrated within policy gradient algorithms such as PPO?
6. Are the performance metrics normalized across tasks to ensure fair comparison?
7. How is the stopping criterion (“Stop”) determined—manually, or via a predefined threshold or rule?
8. What does the statement about activation means collapsing “without empirical constraints” mean in the context of reinforcement learning or policy gradient optimization?

**Details Of Ethics Concerns:**

None.

---

### Meta-Review · Area_Chair_omML · 2025-12-29

**Summary:**

The paper “Imitate Optimal Policy: Prevail and Induce Action Collapse in Policy Gradient” studies Action Collapse in optimal policy networks and proposes a remedy that leverages a simplex ETF structure. The approach consists of (1) characterizing Action Collapse in idealized discrete-action settings, (2) showing that fixing the final-layer weights to an ETF induces this structure, and (3) integrating this constraint into standard policy gradient methods. Experiments on classic control and Atari environments indicate faster convergence and improved stability.

**Reviewer Concerns:**

The paper “Imitate Optimal Policy: Prevail and Induce Action Collapse in Policy Gradient” studies Action Collapse in optimal policy networks and proposes a remedy that leverages a simplex ETF structure. The approach consists of (1) characterizing Action Collapse in idealized discrete-action settings, (2) showing that fixing the final-layer weights to an ETF induces this structure, and (3) integrating this constraint into standard policy gradient methods. Experiments on classic control and Atari environments indicate faster convergence and improved stability.

**Reviewer Scores:**

As no author rebuttal was available for this submission, there was no opportunity for reviewers to revise their assessments.

---

### Decision · Program_Chairs · 2026-01-26

Reject